# Insulin Predicts Methotrexate Response by Affecting the Transcription of Methotrexate Target Genes in the Treatment-Naive Rheumatoid Arthritis

**DOI:** 10.3390/cells14130964

**Published:** 2025-06-24

**Authors:** Victoria M. E. Lundgren, Malin C. Erlandsson, Venkataragavan Chandrasekaran, Sofia Töyrä Silfverswärd, Rille Pullerits, Maria I. Bokarewa

**Affiliations:** 1Department of Rheumatology and Inflammation Research, Institute of Medicine, University of Gothenburg, Box 480, 40530 Gothenburg, Sweden; guslunviv@student.gu.se (V.M.E.L.); malin.erlandsson@rheuma.gu.se (M.C.E.); venkataragavan.chandrasekaran@gu.se (V.C.); rille.pullerits@rheuma.gu.se (R.P.); 2Rheumatology Clinic, Sahlgrenska University Hospital, Gröna stråket 12, 41345 Gothenburg, Sweden; 3Department of Ophthalmology, Sahlgrenska University Hospital, Region Västra Götaland, 41345 Gothenburg, Sweden; sofia.silfversward@neuro.gu.se; 4Department of Clinical Immunology and Transfusion Medicine, Sahlgrenska University Hospital, Region Västra Götaland, 41345 Gothenburg, Sweden

**Keywords:** rheumatoid arthritis, methotrexate, prediction, insulin

## Abstract

Methotrexate (MTX), the most common first-line treatment in rheumatoid arthritis, is often insufficient, with no model capable of predicting response. The RA classification criteria, including autoantibodies and inflammation, were applied to 257 patients with newly diagnosed inflammatory arthritis in the cohort study, estimating MTX response. A total of 172 patients received MTX as the first anti-rheumatic drug and response was recorded at 1 year follow-up. A multivariable logistic regression used variables distinct between MTX-responders and non-responders to build the predictive model of response. Overall, 53.5% of MTX treated patients responded. Non-responders were frequently autoantibody positive, and responders were older, had lower RA classification scores, frequent corticosteroid use, and high insulin levels at baseline. Inflammation parameters were comparable between the groups. In the multiple regression analysis, the RA classification score and age at the first visit were strong predictors of MTX response (AUC 0.697, *p* < 0.0001). Including blood levels of insulin and IFNg improved AUC to 0.782 (*p* < 0.0001), offering early discrimination between responders and non-responders with high accuracy. Cellular experiments showed that insulin could be used to estimate MTX response by demonstrating that insulin changed the transcription of MTX target genes in the folate metabolism after exposing CD4+ cells ex vivo, which could facilitate MTX response in immune cells.

## 1. Introduction

Rheumatoid arthritis (RA) is a typical autoimmune inflammatory disease with increasing functional disability, which causes severe limitations for quality of life and progressively growing healthcare costs for society. The severity and course of the disease vary, from spontaneous resolution of the arthritis to persistently active polyarthritis rapidly progressing to skeletal damage and limited joint functioning. A consolidation of years of clinical experience resulted in the development of the RA classification criteria for early identification of patients with poor prognosis. These criteria combine clinical severity of the disease by enumerating affected joints with systemic inflammation and presence of RA-specific antibodies such as rheumatoid factor (RF) and anti-citrullinated peptides antibodies (ACPA) [1]. Although not meant for diagnostic purposes, the RA classification criteria are frequently used to support diagnosis, unifying the diverse population of patients with inflammatory arthritis.

Modern advances in anti-rheumatic treatment made it clear that early initiated treatment efficiently suppresses inflammation and RA progression correlates to better long-term outcomes including survival, comorbidities, and functional performance [2]. In the broad arsenal of new antirheumatic drugs, methotrexate (MTX) remains the first-choice treatment for majority of RA patients [3]. MTX is a slow acting drug. In general, it takes up to 12 weeks to reach its maximal clinical effect. Sufficient clinical response is generally reached by about 50% of the patients who complete this treatment period [4] while the remaining patients need a combined or a different drug treatment. The loss of time because of non-response to MTX is especially harmful for patients with persistent inflammation and high disease activity. Thus, there is an urgent need to prospectively identify which patients are likely to respond to MTX, making them good candidates for treatment, and inversely, refrain from treating potential non-responders with MTX and instead pivoting early to the use of other drugs to prevent disease progression.

A recent review [5], summarized more than 100 individual parameters and 11 multivariate predictive models for estimation of treatment response based on clinical improvement over the period of 6 months. Many studies reported a high positive prediction power of >0.8 but were unable to replicate the results when tested in an external cohort.

Starting with parameters formally known to correlate with poor RA prognosis [6], demonstrated that information of traditional clinical and laboratory parameters such as antibodies, radiographic damage, age, gender, and symptom duration were insufficient to predict MTX response. Thereafter, the models combined traditional parameters with the genetic markers. Through the genetic approach, the strongest prediction was obtained by the model using genomic loci for IFNAR, IL-17A, CXCL13, IL21R, and IL23 [7]. However, this study was weakened by a small number of patients. Other promising genetic approaches employed single nucleotide polymorphism (SNPs) analysis in the genes of adenosine metabolism [8,9], in the genes of MTX transporters and DNA synthesis [10], and variants of the HLA genes [11]. Finally, the study of Sysoev et al. [12] used all known genetic risk variants associated with RA and found no predictive relation between genetic findings and MTX response. Extending from genetic approaches, metabolomic models were proposed based on protein degradation products [13] and lipidomics [14]. All in all, these approaches have failed to reach a consensus on which parameters are associated with MTX non-response.

Hyperinsulinemia and the ensuing insulin resistance is frequently found in patients with RA [15]. The comparable serum levels of IL-6 in control and RA groups [16] pointed to the role of non-inflammatory mediators in insulin resistance development. Similarly, exploring the predictive potential of different cytokines for MTX response [17] revealed no correlations between interleukins and treatment response to MTX while conflicting results were obtained regarding TNFa. Interestingly, levels of interferon (IFN)a [18] and an IFN-responsive gene signature [19] were demonstrated as important contributors to poor treatment response. The IFN-signaling pathway contributes to the transition from pre-clinical RA to a clinically persistent disease, mainly through driving an aggressive immune response via lymphocyte proliferation, altered cellular metabolism, and differentiation into effector subsets. Recently, it was shown that IFNg production suppressed insulin response through downregulation of the insulin receptor [20]. Inversely, we and others have shown that insulin exposure induces cellular senescence and suppresses IFNg production by peripheral blood leukocytes [21,22]. Other non-inflammatory mediators, like survivin, IL8, and Flt3-Ligand, were also associated with a poor RA outcome [19,23,24,25]. Therefore, it raises the possibility that MTX response can be crucially linked to the pre-existence of these mediators at RA onset, more so than canonical inflammatory molecules.

In this study, we investigated if non-traditional measures of inflammation such as IFNg, IL8, Flt3-ligand, insulin, and survivin combined with the RA classification criteria are useful for estimation of MTX response in patients with early treatment-naïve inflammatory arthritis.

## 2. Materials and Methods

### 2.1. Study Cohort

The inception cohort for this prospective clinical study comprised the patients referred by general practitioners for assessment to the Rheumatology Clinic, at the Sahlgrenska University Hospital of Gothenburg during the periods between 5 November 2012–4 November 2013, and 1 July 2018–30 June 2019. The collection strategy was identical for both time periods. In total, the study comprised 257 patients with no difference with respect to age and gender distribution (Appendix A).

### 2.2. Eligibility Criteria

Inclusion criteria: absence of rheumatological diagnosis at the 1st visit, clinical signs of arthritis, age above 18 years. Exclusion criteria: inability to give informed consent, diagnosis of gout, spondylarthritis, reactive arthritis given at 1st visit, and use of immunosuppressive drugs at the 1st visit to rheumatologist.

### 2.3. Data Collection

Digital medical records of the 1st and all consecutive visits were reviewed for the period of 60 months. Evaluation was performed by two independent researchers. The records were available for all but 4 patients who had no follow-up visits. These patients were excluded from the study. Clinical and treatment information, family history, and smoking habits were collected independently from serological characteristics of the complete blood counts, acute-phase reactants, and RA-specific antibodies.

### 2.4. Clinical Outcomes

For this study, response to treatment within the 1st year was used as the primary outcome. To evaluate MTX response, we used the medical records of the 1 year visit which included swollen and tender joint counts, CRP, and ESR. If the patient received MTX at a later visit, 1 year was calculated from the date the patient was given MTX. To be classified as a MTX responder, the patient needed to continue MTX monotherapy and/or had 2 or less swollen joints and normal levels of CRP and ESR. In cases where the detailed information about joint counts was not available, the assessment made by the treating rheumatologist was considered. If the doctor considered the patients’ medication was sufficient, this judgment was also accepted in our study. Other clinical outcomes at 1 year were the weekly dose of MTX, use of oral corticosteroids, combination of several DMARDs, and the number of tested DMARDs. MTX treatment was started in 171 patients. Three patients who used MTX for the period of less than 1 months were analyzed as non-MTX users. An additional 4 patients who received MTX at a later visit were analyzed in MTX group. Nine patients who started with combination treatment of MTX and other DMARDs (1 sulfasalazine and hydroxychloroquine, 1 sulfasalazine, 3 abatacept; 3 TNF-inhibitors, 1 tocilizumab) at the 1st visit were analyzed as MTX non-responders. Among the patients not treated with MTX (*n* = 85), 17 patients received sulfasalazine, 4 patients received hydroxychloroquine, and 4 received biologics (1 rituximab, 3 TNF-inhibitors). Nineteen patients were treated with oral corticosteroids.

### 2.5. Joint Assessment

Joint assessment at the 1st visit was performed on the 44 swollen joint counts and included the sternoclavicular, acromioclavicular, shoulder, elbow, wrist, metacarpophalangeal, proximal interphalangeal, knee, ankle, and metatarsophalangeal joints. This information was used in the RA classification score assessment and for calculation of DAS28 and DAS44 articular indexes.

### 2.6. RA Classification

The RA classification score was calculated based on the ACR/EULAR consensus criteria [1]: number of swollen joints where 1 large joint, 0 points; 2–10 large joints, 1 point; 1–3 small joints, 2 points; 4–10 small joints, 3 points; and more than 10 joints with at least 1 small, 5 points. Additional points were given for the presence of RA-specific antibodies rheumatoid factor (RF) and antibodies to cyclic citrullinated peptides (ACPA), and elevated levels of erythrocyte sedimentation rate (ESR) and C-reactive protein (CRP). The high levels for both RF and ACPA were defined as the cutoff at three times the upper limit on normal of the routine immunoassay. A low positive RF or ACPA received 2 points; a high positive RF or ACPA received 3 points. A total of 6 points or more indicated definite RA.

### 2.7. Analysis of RA-Specific Autoantibodies

Anti-citrullinated protein antibodies (ACPA) were measured in serum by automated multiplex method (Bioplex2200, Biorad, Hercules, CA, USA) at the accredited Laboratory of Clinical Immunology, Sahlgrenska University Hospital. For ACPA, the cut-off levels above 3.0 U/mL were set positive by the manufacturers and validated on healthy individuals in the laboratory. During the period 2012–2013, total RF was measured by rate nephelometric technology (Beckman Immage 800, Beckman Coulter AB, Brea, CA, USA). The levels of total RF above 20 U/mL were considered positive. During the period 2018–2019, RF of IgM isotype was performed using the EliA immunoassay on the Phadia 250 system according to manufacturer’s instructions (Phadia AB, Thermofisher Scientific, Uppsala, Sweden). Levels above 5 kIE/L were considered positive. The high levels for both RF and ACPA were defined according to the RA classification criteria [1] as cutoff at three times the upper limit on normal. Indication of antibody positivity and high levels of antibodies were used for EULAR score calculation. For the multiple regression analysis, antibodies were ranked from 0 to 3 for positive results of none, any, or both antibodies.

### 2.8. Inflammation Index

The inflammation index was built based on results at the 1st visit, including the inflammatory markers CRP, ESR, white blood cell (WBC), and platelet counts. One point was given for each of the following CRP > 10 mg/mL, ESR > 20 mm/h, WBC > 9 × 10^9^/L, and platelet > 350 × 10^9^/L. The points were then summarized, giving each patient an inflammation index between 0 and 4.

### 2.9. Serological Measures

Insulin levels were measured by reverse sandwich ELISAs (DY8056, R&D Systems, Minneapolis, MN, USA). Cytokine levels were measured by sandwich ELISAs for survivin (DY647), IL-8 (DY208) (both, R&D Systems), and IFNg (M1933) (Sanquin, Amsterdam, The Netherlands). All measures were performed in frozen serum samples diluted 1:2.

### 2.10. Isolation and Stimulation of CD4+ Cells

Human peripheral blood mononuclear cells (PBMC) were isolated from the venous heparinized peripheral blood by density gradient separation on Lymphoprep (Axis-Shield PoC As, Dundee, Scotland). CD4+ cell cultures (1.25 × 10^6^ cells/mL) were prepared from fresh PBMC cultures by positive selection (Invitrogen, Carlsbad, CA, USA, 11331D) in RPMI medium (Gibco, Waltham, MA, USA) containing 50μM β-mercaptoethanol (Gibco, Waltham, MA, USA), Glutamax 2 mM (Gibco), gentamicin 50 μg/mL (Sanofi-Aventis, Paris, France), and 5% fetal bovine serum (Sigma-Aldrich, St. Louis, MO, USA) at 37 °C in a humidified 5% CO_2_ atmosphere. For RNAseq and qPCR, cells were cultured in wells coated with anti-CD3 antibody (0.5 μg/mL; OKT3, Sigma-Aldrich, St. Louis, MO, USA), and treated with insulin (0 and 10 nM, Humalog 100 U/mL, Eli Lilly, Indianapolis, IN, USA) for 72 h.

### 2.11. Transcriptional Sequencing (RNA-Seq)

RNA of CD4+ cells was prepared using the Norgen Total RNA kit (17200 Norgen Biotek, Thorold, ON, Canada). Quality control was performed by Bioanalyzer RNA6000 Pico on Agilent2100 (Agilent, Santa Clara, CA, USA). Deep sequencing was performed by RNA-seq (Hiseq2000, Illumina, San Diego, CA, USA) at the core facility for Bioinformatics and Expression Analysis (Karolinska Institute, Huddinge, Sweden). Raw sequence data were obtained in Bcl-files and converted into fastq text format using the bcl2fastq program from Illumina.

### 2.12. Transcriptome Analysis

The mapping of transcripts was performed using Genome UCSC annotation set for hg38 human genome assembly. Analysis was performed using the core Bioconductor packages in R-studio v. 4.4.1. Differentially expressed genes (DEG) after stimulation of CD4+ cells with insulin between the samples were identified using DESeq2 (v.1.44.0) with Benjamini–Hochberg adjustment for multiple testing. Using the normalized expression data in CD4+ cells of 80 MTX-naïve patients, linear regression of plasma insulin levels to the transcriptome was performed. Using design formula as ~log_Insulin_scaled in the DESeq2 pipeline, we identified the genes that showed significant change (|log2FC| > 0, *p*-value ≤ 0.05) for unit increase in level of insulin.

### 2.13. Statistical Evaluation

The patients were grouped by MTX treatment and further by MTX response and non-response at 1 year follow up. GraphPad prism v.10, and R studio v.4.4.1 were used for the analysis. Data are presented as mean ± SD, median [IQR], or in absolute numbers, as appropriate. Missing data for tender and swollen joints (1 patient, 0.004%) and use of oral corticosteroids (2 patients, 0.078%), insulin levels (3 patients, 1.2%) and IFNg (3 patients, 1.2%) at 1st visit, and MTX dose (8 patients, 3.1%) at 1 year were imputed using R-package missForest. Continuous data were analyzed by the non-parametric Mann–Whitney U test when two groups were compared, or Kruskal–Wallis non-parametric ANOVA when 3 groups were compared, followed by Dunn’s post hoc-test, and Spearman’s correlation test. For comparison of frequencies Chi2-tests were used and presented as mid-exact values (www.openepi.com, accessed on 20 October 2024). Relative risk prediction was performed using multivariate logistic regression analysis. The variance inflation factor was used as a measure of multicollinearity between the parameters. Sensitivity analysis was performed by calculating the area under the receiver operative characteristic (ROC) for each model. The final step of analysis provided variable specific beta estimates along with row prediction estimates for individual patients. All tests were two-tailed and conducted with 95% confidence.

### 2.14. Data Availability

Transcriptome sequencing data of insulin-stimulated CD4+ cells was deposited in NCBI GEO with accession GSE282515. Transcriptome sequencing data of CD4+ cells from MTX-naïve RA patients and healthy controls was deposited in NCBI GEO with accession GSE282517.

### 2.15. Ethical Considerations and Approval

The study protocol was reviewed and approved by the Swedish Ethical Review Authority of the West Götaland Region (Dr.nr 257-13). The study was registered at ClinicalTrials.gov with ID NCT03444623. The study was conducted at the Rheumatology Clinic of the Sahlgrenska University Hospital in agreement with the regulation for clinical research of West Götaland Region and in accordance with relevant Swedish guidelines and regulations and following the Good Clinical Practice. The study was based on the records of routine rheumatologic evaluation and the results of blood analysis after the referral from general practitioners. The informed consent was obtained from all subjects in case of additional clinical visits or blood sampling.

### 2.16. Use of Generative Artificial Intelligence (GenAI)

GenAI has not been used in this paper to generate text, data, or graphics, or to assist in study design, data collection, analysis, or interpretation.

## 3. Results

### 3.1. MTX Was the Drug of Choice in the Treatment-Naïve First Visit Patients with Severe Inflammatory Arthritis

Among the total cohort of 257 patients with newly diagnosed inflammatory arthritis, 172 (67%) patients started with MTX as the first-choice treatment. The MTX treated patients were comparable with the group of non-MTX treated patients with respect to age, gender distribution, and smoking habits (Table 1).

The patients who started MTX treatment had the RA classification score significantly higher compared to non-MTX treated patients (Table 1, Figure 1A). This was largely explained by a higher frequency of RF/ACPA positive patients (48.3% vs. 18.8%), a higher number of swollen/tender joints (mean 6.02 vs. 2.74, Figure 1B), and a higher frequency of CRP/ESR abnormalities (52.9% vs. 70.3%, *p* = 0.0083). This generated a significantly higher total RA classification score of the MTX-treated compared to non-MTX treated patients (Figure 1A).

The absolute levels of inflammation parameters ESR, CRP, WBC, and platelet counts were also significantly higher in MTX-treated patients (Figure 1D–F). Consequently, 68.7% of MTX-treated patients and only 23.5% of non-MTX treated patients reached six points in RA classification score at the first visit and were classified as definite RA. This relatively low frequency of definite RA was partly due to the dominance of seronegative patients lacking both RF and ACPA. Additionally, a substantial proportion of patients (Table 1) had started the use of oral corticosteroids before the first visit, which inevitably affected both the swollen joint counts and inflammatory measures.

### 3.2. Severe Joint Disease Is Linked to a Lack of MTX Response

Among the MTX-treated patients, 92 patients (53.5%) were considered treatment responders and 80 were MTX non-responders (Table 1). MTX responders reached low disease activity and even remission on MTX monotherapy within 1 year. The weekly MTX dose of MTX responders was comparable to that of MTX non-responders. Also, the MTX responders frequently stopped using oral corticosteroids (25% vs. 39%, *p* = 0.070). The use of MTX was stopped in about 25% of patients, equally often among MTX responders and non-responders. MTX non-responders continued using MTX in combination with conventional or biologic DMARD. However, the remission rate among MTX non-responders was low. After 1 year, the remission rate in the group of MTX non-responders was significantly less frequent compared to the MTX responders (26% vs. 55%, *p* = 0.0001) and to the non-MTX treated group (80%, *p* = 0.00052). Analyzing clinical and inflammation measures at the first visit, we found that MTX non-responders had a higher RA classification score (Figure 1A) with higher swollen joint counts (Figure 1B), and a higher frequency of RF and ACPA (Figure 1C), while the absolute levels of inflammation parameters were comparable between MTX responders and MTX-non-responders (Figure 1D–G).

### 3.3. Low Insulin Levels Are Associated with a MTX Non-Response

Intrigued by the lack of difference in the traditional inflammation parameters, we explored serum levels of other biomarkers which have been associated with an unfavorable outcome of RA with respect to erosivity and response to TNF-inhibitors [19,23,24,25]. We found that MTX non-responders tended to have higher serum levels of IFNg (median 5.19 vs. 4.04 pg/mL, *p* = 0.078. Appendix A) and significantly lower levels of insulin (median 375 vs. 306 pmol/L, *p* = 0.0052. Appendix A) compared to MTX responders. Serum levels of IL8, survivin and Flt3-ligand were comparable between the groups (Appendix A).

Since insulin levels could be affected in patients with diabetes mellitus (DM) by external supplementation of insulin or by insulin resistance (Table 1), we compared insulin levels in MTX responders and MTX non-responders after adjustment for DM-free patients and confirmed that the difference remained significant (*p* = 0.0201). Next, we investigated if the blood levels of IFNg and insulin were affected by using oral corticosteroids at the first visit. A comparison of insulin and IFNg levels between the users and non-users of oral corticosteroids showed no significant difference in these parameters (Appendix A), neither in the group of MTX-responders nor in MTX-non-responders.

To investigate the relationship between the RA classification score and individual clinical and serological parameters, we performed Spearman’s correlation analysis. The obtained Spearman’s correlation coefficients were overall low, denoting only a weak relationship between the analyzed parameters, with the exception of CRP and ESR (Appendix A).

### 3.4. Development of Predictive Model for MTX Response

In our study, the RA classification score at the first visit consistently recognized patients with more severe arthritis both in those that started MTX treatment and in those non-responding to MTX. Thus, we applied the multiple logistic regression analysis to study if the RA classification score could be used for early prediction of MTX non-response in treatment-naïve patients.

We found that the high RA classification score appeared to be a strong predictor of MTX response (AUC 0.651, 95%CI 0.569–0.733, *p* = 0.0006. Figure 2A). Adding to the RA classification score, information about age, gender, smoking, DM, and use of oral corticosteroids at the first visit showed that only age and use of oral corticosteroids improved the prediction (PPP 67.4%, NPP 64.9%. AUC 0.743. Appendix A). In contrast, when the individual parameters of the RA classification score were used in the prediction model, only swollen joint counts (*p* = 0.028) and RA-antibody positivity (*p* = 0.013) remained significantly associated with MTX response. The inflammation parameters, tender joint count, gender, and smoking were eliminated from the model (Appendix A).

Among the inflammation parameters seldom used in everyday rheumatology, we tested if the blood levels of IL8, IFNg, insulin, survivin, and Flt3-ligand were associated with MTX response. We found that insulin and IFNg levels appeared to be significantly associated with MTX response (AUC 0.644 (95%CI 0.563–0.726), *p* < 0.0011. Figure 2B). Combining in a model RA classification score, age, gender, blood levels of insulin, and IFNg refined the AUC to 0.762 (95%CI 0.693–0.832), *p* < 0.0001. Figure 2B. The parameters of the final model were quantified for multicollinearity and showed the values of variance inflation factor was close to 1, indicating the lack of collinearity between the parameters of this model (Figure 2C).

Employing the beta-estimates given to each parameter within the model, we calculated individual values predicting MTX response for every patient. Expectedly, MTX responders had significantly higher prediction values compared to non-responders (median 0.665 vs. 0.433, *p* = 3.6^−10^) and allowed for the correct allocation of MTX responder and non-responder groups with the positive prediction power 73.6% and 65%, respectively. Sub-analysis of individual estimation values of MTX response in the MTX-users having RA classification score above and below six points demonstrated that the combination of RA classification score with age, gender, IFNg, and insulin significantly separated MTX responders among the patients of both groups (Figure 2D).

### 3.5. Insulin Levels Secure Robustness of the MTX Response Prediction

The studied patients originated from two independent cohorts recruited during the period 2012–2013 (*n* = 60) and 2018–2019 (*n* = 112). The cohorts showed no difference with respect to frequency of MTX treated patients, smokers, and users of oral corticosteroids at the first visit (Appendix A). Thus, we applied the multivariate regression analysis consisting of RA classification score, age, gender, blood levels of insulin, and IFNg to MTX treated patients in each cohort separately. We found that the model accurately predicted MTX response in both independent cohorts and resulted in comparable areas under the ROC curve (AUC 0.84 for 2012–2013 cohort, and AUC 0.758 for 2018–2019 cohort) (Figure 2B). Consequently, the positive predictive power (79.2 and 71.0, respectively) and negative predictive power (82.9 and 64.9, respectively) were also comparable. Age (*p*-values 0.0101 and 0.109, respectively) and blood insulin levels (*p*-values 0.0088 and 0.028, respectively) remained strongly associated with MTX response prediction in both cohorts. This secured robustness of the prediction model.

### 3.6. Insulin Levels Affect Transcription of MTX Metabolizing Enzymes in CD4+ Cells

MTX is absorbed into the cells through the folate transporter transmembrane solute carrier *SLC19A1*. Intracellular MTX is polyglutamated to its active form by the enzymes *FPGS* and *GGH*. The modified MTX inhibits several enzymes of the folate metabolism, including *MTHFR, MTR, MTRR*, and thymidylate synthetase *TYMS* and *ATIC* that together, regulate the production of purines and pyrimidines required for DNA and protein synthesis, *AMPD2/3* that control adenosine release, and dihydrofolate reductase *DHFR* catalyzing nitric oxide synthase uncoupling (Figure 3A). Genetic variants in the MTX targeting enzymes have been repeatedly associated with poor response to MTX treatment [8,10,26]. In CD4+ cells, the enzymes affected by MTX are highly expressed (Figure 3B). To investigate the effect of insulin on these enzymes, we profiled the transcriptome through RNA-seq after stimulating CD4+ cells with insulin (six healthy controls, 10 nM, 48 h). In parallel, we performed linear regression of gene expression in CD4+ cells to increasing levels of insulin using 80 RNA-seq datasets of MTX treatment-naïve patients.

We found that both ex vivo stimulation of CD4+ cells with insulin and increasing plasma insulin levels induced changes in the folate transporter SLC19A1 and folate processing enzymes MTHFR, MTRR, MTR, and MTHFD. Insulin inhibited the transcription of AMPD2/3, TYMS, and DHFR, contributing to modes of action of MTX intracellularly. Together, these experiments convincingly demonstrate interference of extracellular insulin in transcription of the key enzymes of the folate metabolism. Therefore, we concluded that extracellular plasma insulin could facilitate MTX response in treatment-naïve RA patients.

## 4. Discussion

This study demonstrated the role of blood insulin levels for the estimation of MTX response in the treatment-naïve, newly diagnosed patients with inflammatory arthritis.

Insulin was a strong denominator in the MTX response prediction model in which we utilized the absolute levels of insulin. Insulin levels generally increase during severe infections and chronic inflammation and alter the inflammatory response. Hyperinsulinemia can result from both reduced insulin uptake due to decreased insulin receptor levels and attenuation of insulin signaling pathway by modification the intermediate signaling molecules like the insulin receptor substrates. The temporal progression of arthritis disease raises the inflammatory milieu in RA and desensitizes the response to insulin through both these mechanisms. Insulin resistance assessed through HOMA-IR was associated with tender joints and HAQ score in seropositive RA [27] but not to the DAS score or swollen joints, implying that insulin resistance can arise already in the early phase of the disease, if not prematurely. Regardless of the cohort collection timepoint in our study was during 2012–2013 or 2018–2019, our prediction model highlights that insulin sensitivity and reduced IFNg production marks a positive response to MTX. Therefore, assessing insulin resistance and IFN levels can be informative for adjusting treatment for an early RA patient. If combined with inflammatory parameters, it can be helpful for tracking multi-drug treatment duration and quickly reverting once disease severity is minimized, and a stable insulin sensitive state is reached.

Most of the blood samples in our study were collected before the visit to rheumatologist. This scenario impeded the gathering of information about blood sampling conditions and details of the patients’ metabolic profile other than diabetes mellitus diagnosis at first visit. Hence, the insulin resistant state of the patients could not be evaluated properly and should be indicated as a weakness of this study. To partly compensate for this drawback, we explored the effect of insulin on a transcription profile of CD4+ cells as a function of blood circulating levels and by direct stimulation of cell cultures with insulin ex vivo. We demonstrated that expression of folate transporters, as well as folate processing enzymes in CD4+ cells, was proportionally dependent on blood insulin levels in both analyses. This information provides new insight in insulin’s effect on treatment response in RA and demonstrates its functional impact in non-metabolic condition. Therefore, restoring insulin resistance in RA patients with hyperinsulinemia can be a reasonable intervention point to rescue poor MTX response.

Notably, in prediction of MTX response, insulin levels are disentangled from systemic inflammation, since ESR, CRP, WBC, and platelet counts were eliminated from the predictive model. However, the regulation of insulin levels and the health effects of insulin are manly studied in the metabolic context of obesity and diabetes research. The functional role of insulin outside metabolic conditions and the direct effect on immune cells remains mainly unexplored. Our recent report emphasized anti-inflammatory properties of insulin counteracting IFNg production and the proliferation of CD4+ cells [22]. The inverse relationship between insulin and IFNg levels is strengthened by the model of MTX response prediction proposed in this study.

Our study patients consisted of independent cohorts enrolled within two separate calendar years. Although collected 5 years apart, the initial treatment regimen did not differ between the cohorts and were remarkably similar in proportion of MTX-treated patients and, also, of responders and non-responders.

The model consisting of RA classification score, age, blood levels of insulin, and IFNg had sufficient positive and negative predictive power and was proved useful for both cohorts. When comparing the prediction cohorts, insulin and age were the factors consistently distinct between responders to non-responders. The RA classification score had a strong predictive power in one of the two cohorts. IFNg levels did not reach significant power when cohorts were separated but was significant in the combined material and improved the accuracy of the predicted model. As proper insulin uptake marks MTX response consistently in these two cohorts regardless of other comorbidities like diabetes mellitus, it is tempting to state that this easily measurable parameter can be indexed regularly to follow disease course and dynamically adjust treatment regimen.

Unexpectedly, the use of oral corticosteroids had no significant impact on insulin levels in this study. One of possible explanations to this fact could be a known strong anti-inflammatory effect of oral corticosteroids, which could contribute to lower RA classification score by suppressing swollen joint counts and inflammation of the treated patients.

Our study shows that the RA classification criteria appeared as a useful tool in predicting MTX treatment response. This new application of the classification criteria is in line with its reflection of RA aggressiveness. Due to the ranking approach, the total RA classification score was more sensitive to treatment response than a combination of the individual classification parameters of swollen joints, inflammation, and antibodies.

The outcome level of MTX response is set high in our study. At 1 year, the responders should have low disease activity or remission on MTX monotherapy. While most of the previous models relied on a traditional change in the disease activity index to identify MTX responders, this stringent definition of MTX response embraces the modern goals of RA treatment with respect to inflammation, pain, and quality of life that have significantly advanced in the era of biological drugs.

We would like to highlight that the parameters of our model are easy to apply in any hospital setting. While other models employed clinical and genetic information (SNPs and known genetic loci for MTX-targeted enzymes) to achieve sufficient predictive power, our model encompasses patient metadata such as age and gender, aside from easy and routine laboratory measures of insulin and IFNg, and the rheumatologist-administered stringent disease classification score. Additionally, these data entered into patient records can prove useful in future studies to investigate the effect of comorbidities on treatment response and disease progression. Therefore, our model is both proactive and applicable at the university clinics, in smaller rheumatology units, and by general practitioners.

## 5. Conclusions

In conclusion, our study shows that blood levels of insulin combined with age, RA classification score, and IFNg were predictive of MTX response in early treatment-naïve arthritis patients. Functionally, insulin modulated transcription of the MTX-target genes in the folate metabolism which could facilitate MTX response.

## Figures and Tables

**Figure 1 cells-14-00964-f001:**
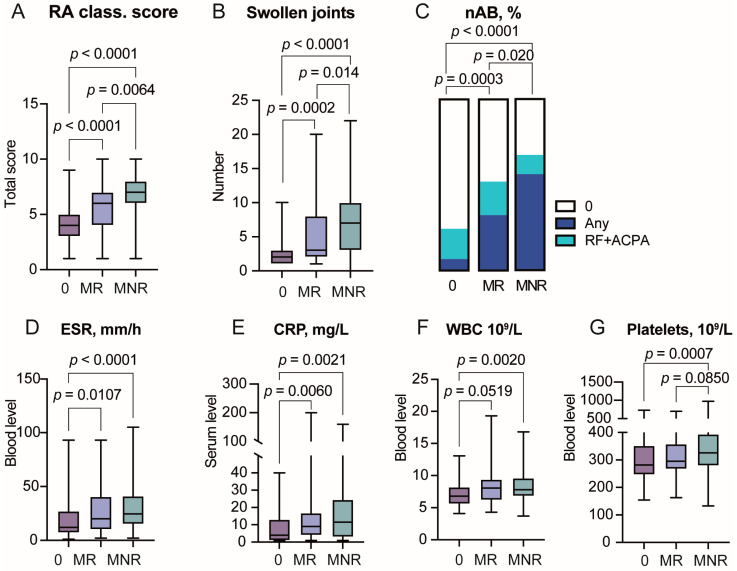
Association between the RA classification score and methotrexate response. (**A**) Box plot of RA classification score in methotrexate-responders (MR, *n* = 92), methotrexate non-responders (MNR, *n* = 80) and patients not treated with methotrexate (0, *n* = 85). (**B**) Box plot of swollen joint counts at the 1st visit. (**C**) Stacked bar plot of frequency of RA-specific antibodies RF and ACPA. AB negative, 0 points; RF or ACPA, 1 point; RF and ACPA, 2 points. (**D**) Erythrocyte sedimentation rate (ESR). (**E**) C-reactive protein (CRP). (**F**) White blood cell (WBC) count. (**G**) Platelet count.

**Figure 2 cells-14-00964-f002:**
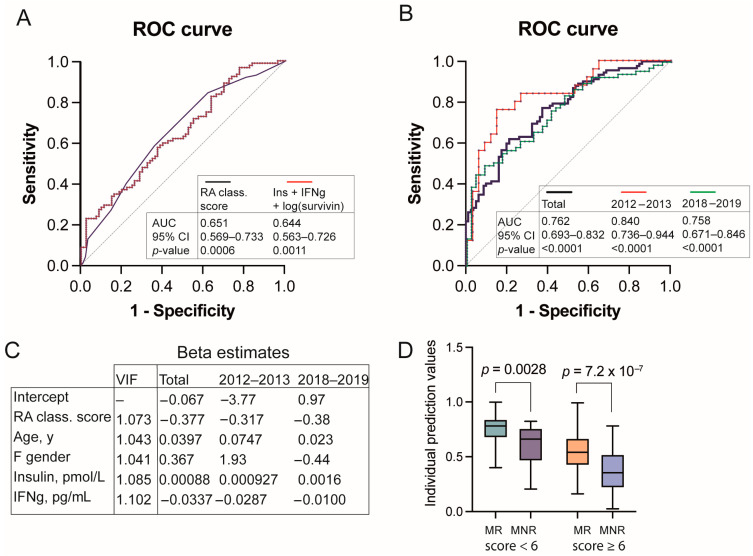
Prediction model of methotrexate response. (**A**) Receiver operating characteristics (ROC) curves of MTX response prediction by RA classification score (black line) and by a combination of blood levels of insulin, IFNg, and log(Survivin) (red line). AUC, area under the ROC curve. CI, confidence interval. (**B**) ROC curves of MTX response prediction by a combination of RA classification score, age, gender, insulin and IFNg. Total cohort (*n* = 172, bold black line); 2012–2013 cohort (*n* = 59, red line), and 2018–2019 cohort (*n* = 113, green line). (**C**) Table of variance inflation factor and beta estimates from the multiple logistic regression model for the total cohort. VIF, variance inflation factor. (**D**) Box plots of individual predictive values for patients with RA classification score above 6 (MR, *n* = 55; MNR, *n* = 62) and below 6 (MR, *n* = 37; MNR, *n* = 18).

**Figure 3 cells-14-00964-f003:**
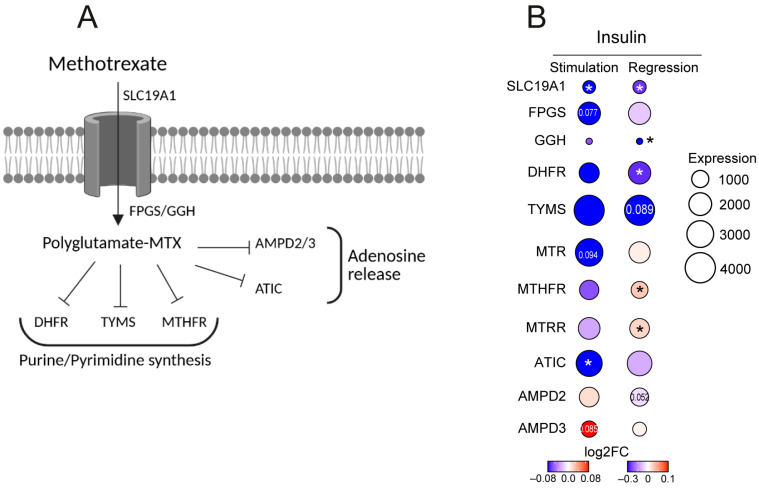
Insulin affects transcription of the methotrexate target genes. (**A**) Cartoon of biochemical reactions regulated by methotrexate (MTX). (**B**) Heatmap of transcription change in genes of MTX target enzymes in CD4+ cells in regression to plasma insulin levels of 80 MTX treatment-naïve patients and in CD4+ cells stimulated with insulin (6 healthy controls, 10 nM, 48 h). SLC19A1, Solute Carrier Family 19 Member 1; FPGS, Folylpolyglutamate synthase; GGH, Gamma-Glutamyl Hydrolase; AMPD2/3, adenosine Monophosphate Deaminase 1; ATIC, Inosine Monophosphate Synthase; TYMS, Thymidylate Synthetase; MTHFR, Methylenetetrahydrofolate Reductase; MTR, Methionine Synthase; MTRR, Methionine Synthase Reductase; DHFR, Dihydrofolate Reductase; MTHFD1, Methylenetetrahydrofolate Dehydrogenase, Cyclohydrolase and Formyltetrahydrofolate Synthetase 1. Numbers in the circles indicate *p*-values. *, *p*-value < 0.05.

**Table 1 cells-14-00964-t001:** Clinical and demographic characteristics of patients with inflammatory arthritis at the 1st visit and at 1 year.

	MTX Responders (*n* = 92)	Non-Responders (*n* = 80)	No MTX (*n* = 85)
Female, *n* (%)	61 (66.3)	55 (68.75)	59 (69.4)
Age, y	58 [22–89]	51 [22–88] *^p^* ^= 0.0045^	52.5 [17–90]
Smokers, *n* (%)	39 (42.4)	29 (36.3)	29 (34.1)
Diabetes mellitus, *n*	10 (10.9)	7 (8.7)	6 (7.06)
RA antibodies, pos	50 (54.3)	56 (70) *^p^* ^= 0.037^	23 (27) *^p^* ^< 0.0001^
RF, *n* (%)	40 (43.5)	49 (61.25) *^p^* ^= 0.021^	17 (20.0) *^p^* ^= 0.00073^
ACPA, *n* (%)	40 (43.5)	52 (65) *^p^* ^= 0.0055^	12 (14.1) *^p^* ^< 0.0001^
RF + ACPA, *n* (%)	30 (32.6)	45 (56.25) *^p^* ^= 0.0020^	6 (7.06) *^p^* ^< 0.0001^
RA classification score	5.85 [1–10]	6.89 [1–10] *^p^* ^= 0.0005^	4.07 [1–9] *^p^* ^< 0.0001^
≥6 points, *n* (%)	55 (59.8)	62 (77.5) *^p^* ^= 0.0059^	20 (23.5) *^p^* ^< 0.0001^
Swollen joints, *n*	5.11 [1–20]	7.06 [0–22]	2.74 [0–10]
Inflammation Index	1.50 (0–4)	1.85(0–4) *^p^* ^= 0.064^	1.13 (0–4) ^*p* = 0.0003^
MTX at 1y, *n* (%)	80 (87)	68 (85)	0
MTX dose at 1y, mg/w	16.4 (0–25)	14.9 (0–25)	0
Other DMARDs, *n*	2 (1.47)	20 (25) *^p^* ^< 0.0001^	14 (16.5)
Biologics, *n*	0 (0)	46 (57.5)	5 (5.9) *^p^* ^= 0.0059^
Tested DMARDs, *n*	1.03 (1–4)	1.84 (1–3) *^p^* ^< 0.0001^	0.29 (0–2) *^p^* ^< 0.0001^
OC at 1st visit, *n* (%)	51 (76)	58 (58) *^p^* ^= 0.022^	23 (27.1) *^p^* ^= 0.0014^
OC at 1 year, *n* (%)	23 (25)	31 (38.75) *^p^* ^= 0.056^	8 (9.4) *^p^* ^= 0.0066^
Remission at 1 y, *n* (%)	51 (55.4)	21 (26.3) *^p^* ^= 0.0001^	68 (80) *^p^* ^= 0.00052^

Data is presented as mean [min–max] and n (%). *p*-values were calculated between group pairs by Mann–Whitney U test or chi-square mid-exact. *p*-values in the column of non-responders refer to comparison with MTX-responders. *p*-values in the column of No MTX refer to comparison with the whole MTX-treated group. MTX, methotrexate, OC, oral corticosteroids.

## Data Availability

Transcriptome sequencing data of insulin-stimulated CD4+ cells is deposited in NCBI GEO with accession GSE282515. Transcriptome sequencing data of CD4+ cells from MTX-naïve RA patients and healthy controls is deposited in NCBI GEO with accession GSE282517.

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
