# Peer review of "Insulin Predicts Methotrexate Response by Affecting the Transcription of Methotrexate Target Genes in the Treatment-Naive Rheumatoid Arthritis"

_cells, 2025, doi:10.3390/cells14130964_

Round 1

Reviewer 1 Report

Comments and Suggestions for Authors

There is no clinically useful model to predict responsiveness to methotrexate treatment in RA. Thus, this work aimed to build a prediction model that is easy to apply in a clinical setting.

This model is based on measuring insulin levels and INFg production in patients' sera, combined with other factors such as age, and the classification score of the disease, and these together can predict MTX responsiveness of early RA patients. They suggest that insulin modulates the transcription of MTX target genes, and this could facilitate the response to MTX.

This is a nice piece of work, and applying this model will shorten the time till a patient can get the right therapy. 

Reviewer 2 Report

Comments and Suggestions for Authors
  • I assume the authors adopted 2010 ACR/EULAR classification criteria for diagnosis of RA, which is not clearly stated in Materials and methods section. It is not clear for me how the patients in Supplementary Table S1 with ‘RA classification score’ being less than 6 were included in the study cohorts. Were the patients described in Subsection 2.1. diagnosed based on 2010 ACR/EULAR classification criteria or patients diagnosed by attending physicians’ own decision included? Or patients diagnosed on 1987 ARA criteria even included?
  • Exclusion criteria: concomitant rheumatic diseases might affect joint manifestations and cytokine profiles. Were the patients with concomitant rheumatic diseases such as secondary Sjögren’s syndrome included or excluded?
  • Line 437: Do the authors think the findings would be applicable to inflammatory arthritis other than RA?

Round 2

Reviewer 2 Report

Comments and Suggestions for Authors

(No further comments)